# Bile Acid Profile and its Changes in Response to Cefoperazone Treatment in MR1 Deficient Mice

**DOI:** 10.3390/metabo10040127

**Published:** 2020-03-26

**Authors:** Jinchun Sun, Zhijun Cao, Ashley D. Smith, Paul E. Carlson Jr, Michael Coryell, Huizhong Chen, Richard D. Beger

**Affiliations:** 1Division of Systems Biology, National Center for Toxicological Research, United States Food and Drug Administration, Jefferson, AR 72079, USA; Zhijun.Cao@fda.hhs.gov (Z.C.); Richard.Beger@fda.hhs.gov (R.D.B.); 2Laboratory of Mucosal Pathogens and Cellular Immunology, Division of Bacterial, Parasitic, and Allergenic Products, Office of Vaccines Research and Review, Center for Biologics Evaluation and Research, United States Food and Drug Administration, Silver Spring, MD 20993, USA; ashdawnsmith@gmail.com (A.D.S.); Michael.Coryell@fda.hhs.gov (M.C.); 3Division of Microbiology, National Center for Toxicological Research, United States Food and Drug Administration, Jefferson, AR 72079, USA; huizhong.chen@fda.hhs.gov

**Keywords:** MR1^−/−^, bile acids, riboflavin, metabolomics, microbiome

## Abstract

Mucosal associated invariant T-cells (MAIT cells) are activated following recognition of bacterial antigens (riboflavin intermediates) presented on major histocompatibility complex class I-related molecule (MR1). Our previous study showed that MR1^−/−^ knock-out (KO) mice (lacking MAIT cells) harbor a unique microbiota that is resistant to antibiotic disruption and *Clostridioides difficile* colonization. While we have characterized the microbiota of this mouse strain, changes in global metabolic activity in these KO mice have not been assessed. Here, LC/MS-based untargeted metabolomics was applied to investigate the differences in the metabolome, specifically in the bile acid (BA) profile of wild-type (WT) and MR1^−/−^ KO mice, as well as how antibiotics change these profiles. BA changes were evaluated in the intestinal content, cecum content, and stool samples from MR1^−/−^ mice and WT mice treated with cefoperazone (Cef). Fecal pellets were collected daily and both intestinal and cecal contents were harvested at predetermined endpoints on day 0 (D0), day 1 (D1), day 3 (D3), and day 5 (D5). KO mice exhibited no changes in 6-hydroxymethyl-8-D-ribityllumazine (rRL-6-CH_2_OH; an MR1-restricted riboflavin derivative) in the stool samples at either time point vs. D0, while WT mice showed significant decreases in rRL-6-CH_2_OH in the stool samples on all treatment days vs. D0. Metabolomics analysis from cecal and stool samples showed that KO mice had more total BA intensity (KO/WT = ~1.7 and ~3.3 fold higher) than that from WT mice prior to Cef treatment, while the fold change difference (KO/WT = ~4.5 and ~4.4 fold) increased after five days of Cef treatment. Both KO and WT mice showed decreases in total BA intensity in response to Cef treatment, however, less dramatic decreases were present in KO vs. WT mice. Increases in taurocholic acid (TCA) intensity and decreases in deoxycholic acid (DCA) intensity in the stool samples from WT mice were associated with the depletion of certain gut bacteria, which was consistent with the previously reported microbiome data. Furthermore, the non-detected TCA and relatively higher DCA intensity in the KO mice might be related to *Clostridioides difficile* infection resistance, although this needs further investigation.

## 1. Introduction

Major histocompatibility complex class I-related molecule (MR1) is able to bind to bacteria-synthesized riboflavin metabolites (antigens), which can be presented to mucosal associated invariant T cells (MAIT) for activation [1,2]. MAIT cells, which are present in high abundance in the gastrointestinal mucosa, defend against some microbial activity and infection [3]. MAIT cells are rapidly activated following MR1 presentation of foreign antigens produced by a wide range of microorganisms, including diverse strains of bacteria and yeast. MAIT cells depend on MR1 and certain pathogens for MAIT cell activation to produce inflammatory cytokines and an array of antimicrobial responses [4,5,6]. To date, no research has been published to study the metabolic changes in the host with or without MAIT cells in response to altered gut microbiota by antibiotic treatment.

MR1^−/−^ C57BL/6 knock-out (KO) mice lack detectable MAIT cells and have been used to investigate the roles of MAIT cells in a range of diseases and conditions. In the pathogenesis of arthritis, the severity of collagen-induced arthritis is reduced in KO mice, while reconstitution with MAIT cells induced severe arthritis [7]. Further, MAIT cells have been reported to play important roles in response to pulmonary infection by organisms including *Francisella tularensis*, *Mycobacterium tuberculosis*, and influenza [8,9,10]. However, the interactions between MAIT cells and the gastrointestinal microbiota remains unknown. Our previous study showed the fecal microbiota of KO mice is distinct from wild-type (WT) mice. MR1^−/−^ mice are resistant to *Clostridioides difficile* infection (CDI) even following antibiotic treatment, and CDI resistance is transferrable when the microbiome of the KO mice is transferred to WT mice via fecal microbiota transplantation [11,12].

Numerous researchers have reported that the metabolic activities of the gut microbiota related to salvaging energy and absorbing nutrients play an important role in host physiology and pathology [13,14]. Bile acids (BA) are a class of host microbiome co-metabolites, which are important for lipid digestion and absorption, host metabolic regulation, and pathogen colonization of the gut. BA hydrolysis, dehydroxylation, and deconjugation reactions involve a broad spectrum of intestinal anaerobic bacteria [15]. Among them, BA 7*α*/*β*-dehydroxylation is carried out by several specific intestinal bacteria (a small fraction of the total colonic flora), including *Bacteroides fragilis*, *Bacteroides thetaiotaomicron*, *Clostridium scindens*, *Clostridium sordellii*, and *Escherichia coli* [15]. Our previous LC/MS-based metabolomics analysis investigated the effects of the antibiotic penicillin on the gut microbiota and the host. The host–microbial interactions were also examined by measuring changes in host gut microbiota co-metabolites [16]. Results showed that antibiotic administration altered the gastrointestinal microbiota, and the altered gut microbiota influence the host metabolome, including co-metabolites (BAs, indole- and phenyl-containing metabolites, amino acids, vitamins, and nucleotides) [16].

Here, LC/MS-based untargeted metabolomics was applied to investigate the effects of antibiotic treatment on the metabolome of MR1^−/−^ KO and WT mice (specifically changes in the composition and overall levels of BAs in the gut). BA changes were evaluated in the intestinal content, cecum content and stool samples from MR1^−/−^ KO mice and WT mice treated with cefoperazone (Cef), an antibiotic with a broad spectrum of activity.

## 2. Results

### 2.1. Riboflavin and Bacteria-Biosynthesized Riboflavin Metabolite Intensity Levels

It has been reported that 6-hydroxymethyl-8-D-ribityllumazine (rRL-6-CH_2_OH) is an MR1-restricted riboflavin derivative (derived from the bacterial riboflavin biosynthesis pathway), which specifically and potently activates MAIT cells [2]. Therefore, the levels of riboflavin and rRL-6-CH_2_OH were quantified in the small intestine, cecal content, and stool samples (Figure 1). No significant changes in riboflavin were observed in the intestinal content, cecal content, or stool samples from the KO mice after Cef treatment vs. day 0 (D0). In contrast, WT mice showed significant decreases in riboflavin present in the cecal content at D1, D3 and D5, and showed significant decreases in the stool samples at all time points. No changes in riboflavin were observed in the intestinal content from WT mice. No rRL-6-CH_2_OH was detected in either the intestinal or cecal contents. In the stool, KO mice had no changes in rRL-6-CH_2_OH at any time points vs. D0; however, WT mice showed significant decreases in rRL-6-CH_2_OH in the stool samples at all time points vs. D0. The decreases in rRL-6-CH_2_OH might be related to the decreases in bacterial population in the WT mice after Cef treatment.

### 2.2. Total Measured BA Intensity Levels in the Intestinal Content, Cecal Content, and Stool Samples from MR1^−/−^ KO and WT Mice

The total BA intensity present in the intestinal content, cecal content and stool was calculated by summing the individual BA intensity data (Figure 2, Appendix A). In terms of fold changes, the total cecal BA intensity of KO mice was ~1.7 fold (Appendix A, Intensity(KO)/Intensity(WT) = 1264.27/742.32 = 1.70) and ~4.5 (Appendix A, 497.30/110.20 = 4.5) fold greater than those from WT mice before (D0) and after Cef treatment (D5), respectively. After Cef treatment in KO mice, the total cecal BA intensity at D0 was ~4.5, 1.9 and 2.5 fold of that at D1, D3 and D5, respectively. In contrast, in WT mice, the total cecal BA intensity at D0 was ~0.2, 0.5 and 6.7 fold of those at D1, D3 and D5, respectively. The total stool BA intensity of KO mice was ~3.3 fold and ~4.4 fold of those in WT mice before (D0) and after Cef treatment (D5), respectively. After Cef treatment in KO mice, the total stool BA intensity at D0 was ~5.0, 7.7, 4.0, 6.15 and 4.2 fold of those at D1, D2, D3, D4 and D5, respectively. In contrast, in WT mice, the total stool BA intensity at D0 was ~6.2, 6.1, 5.3, 5.7 and 5.6 fold of those at D1, D2, D3, D4 and D5, respectively.

### 2.3. Individual BA Intensity Levels in the Intestinal Content, Cecal Content, and Stool Samples from MR1^−/−^ KO and WT Mice

It is well known that antibiotic usage can change gut microbiota content which will ultimately change the functional capability of the gut metabolome [17,18] and the BA profiles [15,17,19]. The primary BAs present in mice are cholic acid (CA) and α- and β-muricholic acid (αMCA and βMCA), which can be conjugated with taurine and glycine. The majority of the primary BAs (>95%) are reabsorbed back into the hepatic system. Primary BAs that reach the large intestine can be biotransformed by the gut microbiota to secondary BAs such as hyocholate (HCA), ursocholate (UCA), lithocholate (LCA), and deoxycholate (DCA), or even to the tertiary BA ursodeoxycholate (UDCA). Therefore, BA levels were profiled and compared (Appendix A) between KO and WT mice. Appendix A displays the heat maps of the log_e_ transformed normalized intensity of BAs in the intestinal content, cecal content, and stool samples.

In the intestinal content, the baseline BA profiles at D0 in KO mice was different from that in WT mice (Appendix A). Specifically, UDCA and DCA were significantly lower than those in WT mice. In contrast, taurohyocholate (THCA, a taurine-conjugated BA) was significantly higher in KO vs. WT mice (Appendix A). In general, non-conjugated BAs were lower in KO vs. WT mice, while taurine-conjugated BAs were higher in KO mice at D0 (although not all differences were statistically significant). These results suggested that the bacteria responsible for taurine deconjugation in the intestinal content from KO mice might be lower vs. WT mice at D0. The BA profile changes caused by Cef treatment in KO mice were different from WT mice (Appendix A). No significant changes were observed in the primary BAs, including both free BAs and conjugated forms, while TUCA and TLCA were significantly reduced in the KO mice after five days of Cef treatment vs. D0. In contrast, 15 out of 18 detected BAs were significantly reduced in the WT mice at D5 vs. D0, while only TCA significantly increased. In general, more dramatic changes in BAs were observed in WT mice in response to Cef treatment compared to KO mice (Appendix A).

In the cecal content, secondary BA intensities, including UCD and HDCA, were significantly higher in KO vs. WT mice at baseline levels (Appendix A). In general, conjugated BAs were less abundant than non-conjugated forms in both KO and WT mice. This result indicated that deconjugation reactions mediated by the microbiota occurred at the cecum. Compared to D0, BAs, including βMCA, CDCA, αMCA, UCA, DCA, UDCA, HDCA and ACA, were decreased at D5 of Cef treatment in both KO and WT mice, but with a smaller reduction in the KO mice. Indeed, some secondary BAs (including DCA, UCA, UDCA, HDCA and ACA) were undetected or close to the limit of detection (LOD) in WT mice at D5. Similar to the intestinal BA changes, more dramatic changes in BAs were observed in WT mice in response to Cef treatment compared to KO mice (Appendix A).

At baseline, all primary (except CA) and secondary BAs (except αMCA) were significantly higher in the stool of KO mice than WT mice (Appendix A). In response to Cef treatment, βMCA, CDCA, αMCA, DCA, UDCA, and HDCA were significantly decreased at all time points after D0 in stool samples from both KO and WT mice. However, KO mice decreased to a lesser extent.

In contrast, CDCA, DCA, UDCA and HDCA were undetected or close to the LOD after one day of Cef treatment and remained so for the duration of collection. TCA and sulfolithcholylglycine were not detected in any stool samples from KO mice. TCA significantly increased at all time points in the stool samples from WT mice.

Intestinal TCA significantly increased at D5 in WT mice, while no significant changes occurred in KO mice. TCA was not the major form present in the cecal content or the stool samples where the deconjugation of TCA to CA by the cecum microbiota occurred; therefore, TCA was undetected in the stool samples from KO mice (Figure 3). For WT mice, TCA significantly increased at D1, D2, D3, D4 and D5 in the stool samples vs. D0. WT mice had significant increases in TCA at D1, D3 and D5 vs. D0 in the cecal content (Figure 3). The intestinal βMCA (one of the major primary BAs) was not significantly changed in KO mice, while it was significantly decreased at D3 in WT mice (Figure 3). Cecal βMCA was significantly decreased at D1, D3 and D5 in KO mice, and at D3 and D5 in WT mice. βMCA was significantly decreased at all time points in stool samples from both KO and WT mice, albeit the KO mice had less reduction. Intestinal DCA (one of the major secondary BAs) was not significantly changed in KO mice, while it was significantly decreased at D1, D3 and D5 in WT mice (Figure 3). DCA in the cecum and stool was significantly decreased at all time points for both KO and WT mice (Figure 3). Intestinal UDCA (a secondary BA) was not significantly changed in KO mice, while it was significantly decreased at D3 and D5 in WT mice (Figure 3). Cecal UDCA was significantly decreased at D1, D3 and D5 in WT mice. Stool UDCA was significantly decreased at all time points for both KO and WT mice (Figure 3), albeit the KO mice had less reduction.

### 2.4. Free Taurine Levels in the Intestinal Content, Cecal Content, and Stool Samples

Changes in the levels of free taurine were linked to the levels of taurine-conjugated BAs and diminished unconjugated BAs in response to the attenuation of the gut microbiota due to Cef treatment in both KO and WT mice (Appendix A). Free taurine intensity levels were significantly decreased at all time points vs. D0 in the stool samples from both KO and WT mice (Figure 4). No significant changes in free taurine levels were observed in either the intestinal or cecal contents.

## 3. Discussion

The schematic plot in Figure 5 summarizes the interconnections of previous microbiome studies and the current metabolomics study. MR1 and bacterial derived riboflavin derivatives are required for the activation of MAIT cells, which conduct antimicrobial activity. Our previous study [11,12] showed that MR1^−/−^ mice had inherent differences in microbial composition when compared to WT mice both before and after Cef exposure. Surprisingly, MR1^−/−^ mice exhibited resistance to CDI, which can be transferrable when the microbiome of MR1^−/−^ mice is transferred to WT mice via fecal microbiota transplantation. Results from our current study and many other studies have reported that *Clostridioides difficile* germination and colonization are influenced by antibiotic-induced shifts in BA metabolism in the gut microbiome [18,20,21]. Further, some studies [15,22] have reported that the oxidation of BA hydroxy groups and deconjugation reactions are carried out by gut microbiota. In contrast, BAs influence gut-associated inflammation by regulating gut mucosal immune cells and modulating the balance of T helper cells that express IL-17a and regulatory T cells [23]. However, the BA profiles and subsequent antibiotic-induced changes have not been assessed in a host lacking MAIT cells. The goal of this pilot study was to examine the differences in BA baseline profiles and BA profile changes in response to antibiotic treatment in KO mice compared to WT mice.

In order to assure the quality of the metabolomics data, the retention time shifts and associated mass accuracies were monitored during the run. The retention time shifts of the 40 compounds present in the synthetic quality control (QC) (covering the whole range of the chromatography) were within 0.02 min, while the mass accuracy of the 40 compounds were ≤ 6 ppm (data not shown). The relative standard intensity variations of the ion features across the QC pooled sample runs ≤ 30% were included for the metabolomics analysis. It must be noted that other changed metabolites in response to the antibiotic treatment in the MR1 KO vs. WT mice will be published separately. These changed metabolites include fatty acids, lipids, amino acids, di-peptides, vitamins, bacteria-related metabolites, nucleotides, metabolites involved in energy pathways, and metabolites involved in oxidative stress pathways (will be published separately).

It has been reported that exposure to Cef via drinking water disrupts the gut microbiota of mice, decreases bacterial load by three orders of magnitude, and alters community composition [24]. Our previous study showed that the stool microbiota of MR1^−/−^ mice were minimally perturbed by Cef treatment when compared with WT mice [12]. In order to investigate whether antibiotic-induced changes in BA metabolism follow a similar pattern, the total BA intensity levels in cecal content and stool samples from both KO and WT mice were measured and compared. Here, the total BA intensity levels of KO mice were higher than those of WT mice both prior to and after Cef exposure (Figure 2). The temporal changes of the total BA intensity and BA profiles in response to the antibiotic treatment in KO mice were distinct from WT mice (Figure 2, Appendix A). These results were similar to the changing microbiota patterns previously observed in Cef-treated KO mice [12]. KO and WT mice had a similar decreasing trend after Cef treatment, albeit with less reduction in KO mice. The decreases in BA intensity level may have been linked to the gut microbiota depletion in the stool samples after antibiotic treatment, which we previously reported [22].

The levels of riboflavin and rRL-6-CH_2_OH (produced by the gut microbiota and responsible for MAIT cell activation) [2] were measured in the intestinal content, cecal content, and stool samples (Figure 1). Significant decreases in riboflavin were only observed in the cecal content and stool samples from WT mice after Cef treatment, but not in KO mice. Decreased riboflavin in WT mice after antibiotic treatment might be due to the more efficient absorption from the cecal content into the bloodstream after further microbiota depletion [25] following Cef treatment vs. KO mice. Significant decreases in rRL-6-CH_2_OH in the stool samples from WT mice could be due to the decreasing numbers of gut bacteria (for instance, *Salmonella* [12]) involved in producing rRL-6-CH_2_OH after Cef treatment. KO mice had a similar decreasing trend of rRL-6-CH_2_OH after Cef treatment, but to a lesser extent. Collectively, the data suggested that the Cef treatment caused a depletion of gut microbiota-dependent riboflavin synthesis in both KO and WT mice, but less severely in KO mice.

Increases in TCA (one of the major taurine-conjugated primary BAs) in the cecal content and stool samples from WT mice after Cef treatment (Figure 3) could be related to the attenuation of the gut microbiota. The TCA results indicated that deconjugation reactions mediated by gut microbiota were depleted due to the antibiotic treatment in WT mice. Previously, decreases in *Lactobacillaceae* were observed in WT stool samples after Cef treatment [12]. *Lactobacillaceae,* including *Lactobacillus johnsonii* 100-100, *L*. *plantarum* 80, and *L*. *acidophilus* [15], are involved in such deconjugation reactions. Furthermore, decreases in free taurine levels in the stool samples from WT mice (Figure 4) were consistent with increases in TCA with consideration that more taurine was conjugated to CA in the form of TCA, causing less free taurine to be present in the system. Interestingly, no TCA was detected in the stool samples from KO mice at all time points. The reasons for this might be that the KO mouse microbiota remained intact enough after Cef treatment that the taurine-conjugated BAs were fully hydrolyzed before fecal excretion. Decreases in DCA (one of the major secondary BAs) in the stool samples from both KO and WT mice after Cef treatment (Figure 3) indicated that the gut microbiota involved in the dehydroxylation reactions were depleted due to the antibiotic treatment. Indeed, decreases in *Clostridiales* (i.e., *C. scindens,* which is involved in dehydroxylation reactions in the large intestine [15]) were observed in WT stool samples after Cef treatment [12]. Similar results in DCA changes and microbiome changes were observed in KO mice, albeit in a less pronounced manner. It has been reported that the higher levels of TCA and lower levels of DCA after antibiotic treatment are linked to the promotion of *Clostridioides difficile* (CD) germination and growth [18,20,21]. In this study, non-detected cecal and stool TCA and relatively more DCA in KO vs. WT mice might be a contributing factor for CDI resistance, which has been observed in our previous studies [11]. Further studies are necessary to determine this mechanism in better detail.

## 4. Material and Methods

### 4.1. Chemicals

Optima LC/MS grade acetonitrile and water were purchased from Fisher (Pittsburgh, PA, USA). BA, riboflavin and taurine standards were obtained from Sigma-Aldrich (St. Louis, MO, USA). Cef was obtained from MP Biomedicals (Santa Ana, CA, USA).

### 4.2. Animal Care and Treatment

All animal experiments were previously described [12]. In brief, the animal experiments were conducted at the Center for Biologics Evaluation and Research, United States Food and Drug Administration (FDA), and were reviewed and approved by the FDA Institutional Animal Care and Use Committee (Protocol #2015-08, approved on 24 May 2018). Adult C57BL/6J or MR1^−/−^ C57BL/6J mice were administered water or 0.5 mg/mL Cef ad libitum in sterile drinking water for one, two, three, four or five days, which was refreshed every other day. Fecal pellets were collected daily throughout the duration of the experiments (*n* = 11, 8, 6, 6, 3, and 3 at D0, D1, D2, D3, D4, and D5, respectively, for both KO and WT groups). After collection, the stool samples were immediately stored at −80 °C for sequencing and metabolomics analysis. Two to three mice from the KO or WT group (control) were sacrificed at D0, D1, D3 and D5. At sacrifice, the small intestinal content and cecum content were collected (*n* = 3, 2, 3, and 3 at D0, D1, D3, and D5, respectively for both KO and WT groups). After collection, intestinal and cecal contents were immediately snap frozen, then stored at −80 °C for metabolomics analysis.

### 4.3. Quality Control in Metabolomics

A quality control (QC) sample, comprised of 40 common chemicals for LC/MS open profiling, was evaluated [26]. QC data were acquired from every 10 sample runs for pooled samples to monitor analytical equipment variability and for data filtering.

### 4.4. Open Metabolic Profiling by UPLC/QTof-MS

Intestinal and cecal contents (~250 mg) were mixed with extraction solvent (1:1 MeOH:water) at a ratio of 1:5 (w/v), or a ratio of 1:10 (w/v) for stool pellets. The mixture was vortexed for 2 min, followed by a 10 min sonication for intestinal and cecal contents or a 15 min sonication for stool samples. After centrifugation at 20,000 *g* for 15 min at 4 °C, the supernatant was then transferred to autosampler vials for the metabolomics analysis.

A 5 µL aliquot of the extracted supernatant was introduced into a Waters Acquity Ultra Performance Liquid Chromatography (UPLC) system (Waters, Milford, MA, USA) equipped with a Waters bridged ethyl hybrid (BEH) C8 column with dimensions of 2.1 mm × 10 cm and a particle size of 1.7 µm. The column was held at 40 °C. The UPLC mobile phase consisted of 0.1% formic acid in water (solution A) and 0.1% formic acid in acetonitrile (solution B). While maintaining a constant flow rate of 0.4 mL/min, the metabolites were eluted using linear gradients of 2–80% solution B from 0 to 15 min, and 80–98% solution B from 15 to 17 min. The final gradient composition was held constant for 2 min, followed by a return to 2% solution B at 19.1 min. Mass spectrometric data were collected with a Waters QTof Premier mass spectrometer (Waters, Milford, MA, USA) operated in positive and negative ionization electrospray modes, as reported previously [16]. Briefly, MS^E^ (at low collision energy to collect precursor ion information, at high collision energy to obtain full-scan accurate mass fragmentation information) analysis was performed on a QTof mass spectrometer set up with 5 eV for low collision energy and a ramp collision energy ranging from 20 to 30 eV. Full scan mode from *m*/*z* 80 to 1000 and from 0 to 22 min was used for data collection. Raw UPLC/MS data were analyzed using Micromass MarkerLynx XS Application Version 4.1 (Waters, Milford, MA, USA) with extended statistical tools. The same parameter settings for peak extraction from the raw data were used as previously reported [27,28]. The aligned data from MarkerLynx analysis for QTof-MS data was filtered using the pooled QC samples based on the following criteria: (i) ions with % relative standard deviation (RSD) less than 30% in the pooled QC samples were included; (ii) ions present in ≥70% of QC samples were included. The identity of compounds was based on the combined information of accurate mass measurement, fragmentation mass spectra, and compared data from a free online database [29,30]. The detected BAs riboflavin and taurine were confirmed by commercially available standards. The identity of rRL-6-CH_2_OH was confirmed by the accurate mass and the fragmentation spectrum as published by others [2].

### 4.5. Statistics

In the metabolome data analyses, the missing values were replaced with half of the minimum value of all samples for each metabolite. A fixed-effects linear regression model was used for statistical analysis of the log_e_ transformed intensity of each metabolite from intestinal and cecal samples. Since multiple samples from the same animal were collected at different time points, a mixed-effects linear regression model was used for statistical analysis of the log_e_ transformed intensity of each metabolite from stool samples. The effect of Cef treatment was estimated by contrasting data at D1, D2, D3, D4 or D5 with data at D0. The effect of the KO was estimated by contrasting data from KO mice with data from WT mice. The Benjamini and Hochberg method was used to calculate the false discovery rate (FDR) across the analytes [31]. The data visualization and statistical analysis were performed using the program R [32].

## 5. Conclusions

LC/MS-based untargeted metabolomics was applied to study BA profiles before and after antibiotic treatment of MR1^−/−^ mice vs. WT mice. Metabolomics data showed that the cecal and stool samples from KO mice had more total BAs (KO/WT = ~1.7 and ~3.3 fold) prior to Cef treatment, and an even higher total BA fold change (KO/WT = ~4.5 and ~4.4 fold) after Cef treatment. Both KO and WT mice showed decreases in total BA intensities in response to Cef treatment; however, less dramatic decreases were present in KO vs. WT mice. Increases in TCA and decreases in DCA in the stool samples from WT mice were commensurate with the depletion of certain gut microbiota, which was consistent with the microbiome data. Furthermore, the non-detected TCA and relative higher DCA in stool samples from KO mice might be a contributing factor in CDI resistance, although this needs further investigation. It must be noted that this study has limitations and more mice are needed in future studies for improved statistics. Additionally, this study does not show a direct role of MAIT cells in altering the microbiota and/or metabolome, but rather reports the differences observed in these KO mice.

## Figures and Tables

**Figure 1 metabolites-10-00127-f001:**
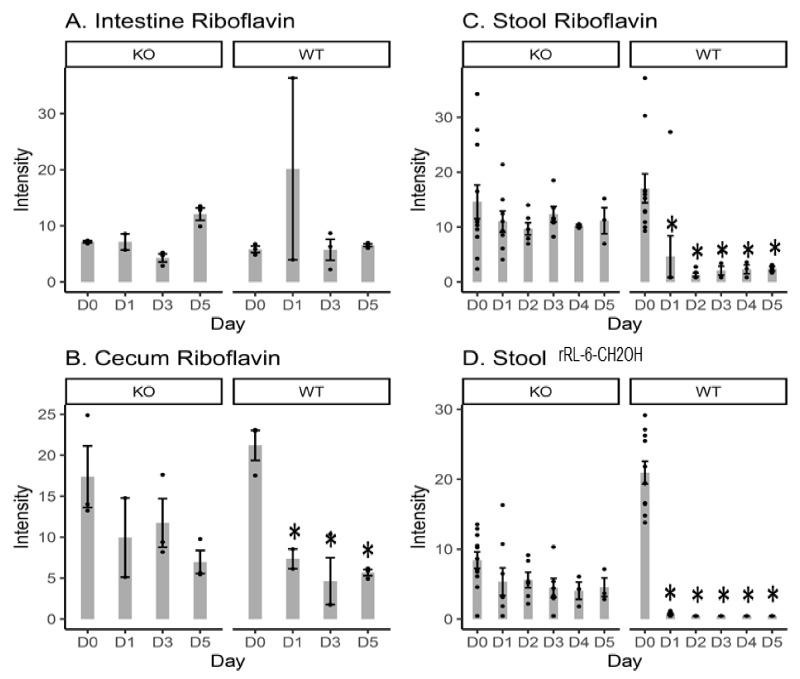
Bar plots of the intensity levels of riboflavin detected in the intestinal content (**A**), cecal content (**B**), and stool samples (**C**); and the intensity levels of 6-hydroxymethyl-8-D-ribityllumazine (rRL-6-CH_2_OH; a major histocompatibility complex class I-related molecule (MR1)-restricted riboflavin derivative) in the stool samples (**D**) from knockout (KO) and wild-type (WT) mice. D0 = day 0 (prior to Cef treatment); D1, D2, D3, D4 and D5 = 1, 2, 3, 4, or 5 days after Cef treatment, respectively; * = significant changes at *p* < 0.05; ● = intensity data from each individual animal.

**Figure 2 metabolites-10-00127-f002:**
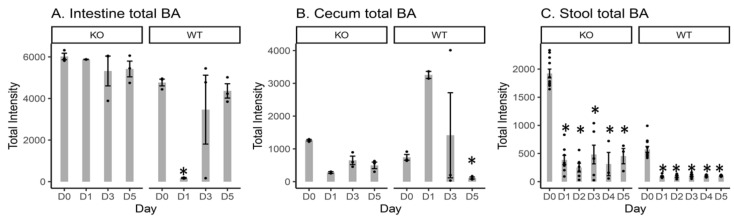
Bar plots of the total bile acid (BA) intensity at D0, D1, D3 and D5 in the intestinal content (**A**) and cecal content (**B**); and at D0, D1, D2, D3, D4 and D5 in the stool samples (**C**) from KO and WT mice. D0 = day 0 (prior to Cef treatment); D1, D2, D3, D4 and D5 = 1, 2, 3, 4, or 5 days after Cef treatment, respectively; * = significant changes at *p* < 0.05; ● = intensity data from each individual animal.

**Figure 3 metabolites-10-00127-f003:**
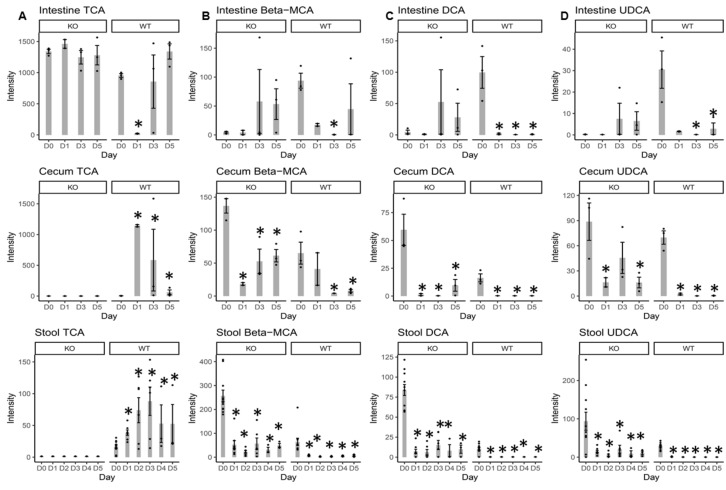
Bar plots of the intensity of primary bile acids (BAs) including taurocholic acid (TCA) (**A**), β-muricholic acid (βMCA) (**B**), and secondary BAs including deoxycholate (DCA) (**C**) and ursodeoxycholate (UDCA) (**D**) in the intestinal content, cecal content, and stool samples from both KO and WT mice. D0 = day 0 (prior to Cef treatment); D1, D2, D3, D4 and D5 = 1, 2, 3, 4, or 5 days after Cef treatment, respectively; * = significant changes at *p* < 0.05; ● = intensity data from each individual animal.

**Figure 4 metabolites-10-00127-f004:**
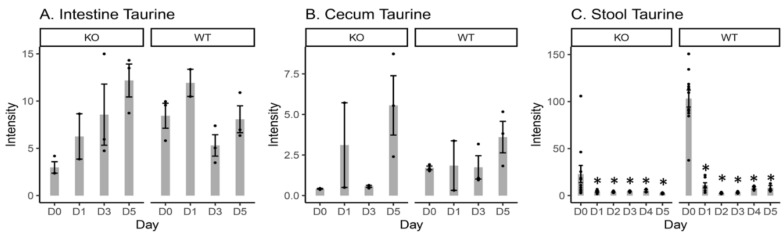
Bar plots of the intensity of free taurine in the intestinal content (**A**), cecal content (**B**), and stool samples (**C**) from both KO and WT mice. D0 = day 0 (prior to Cef treatment); D1, D2, D3, D4 and D5 = 1, 2, 3, 4, or 5 days after Cef treatment, respectively; * = significant changes at *p* < 0.05; ● = intensity data from each individual animal.

**Figure 5 metabolites-10-00127-f005:**
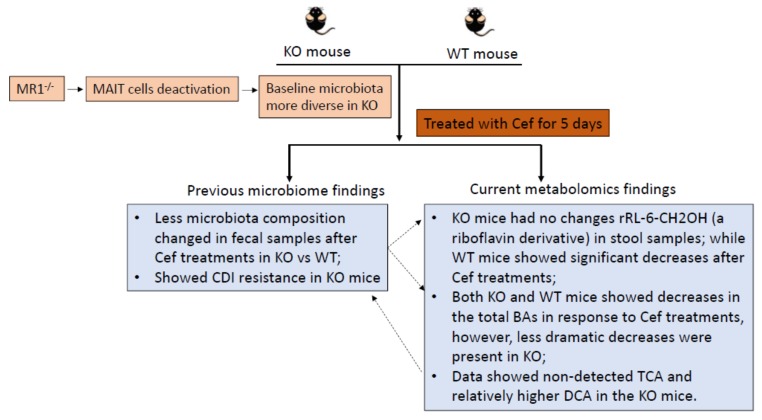
Schematic plot of the interconnections of previous microbiome studies and the current metabolomics study. KO mice with a different microbial composition both before and after Cef treatment might cause the distinct BA profile changes vs. WT mice. In return, the levels of TCA and DCA might influence microbial composition as well. CDI = *Clostridioides difficile* infection.

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
