# Peer review of "Bile Acid Profile and its Changes in Response to Cefoperazone Treatment in MR1 Deficient Mice"

_metabolites, 2020, doi:10.3390/metabo10040127_

Round 1
Reviewer 1 Report
Peer review of Sun et al, Bile acid profiles and its changes in response to Cefoperazone treatment in MR1 deficient mice.
This manuscript uses metabolomics measurements on WT and KO mice missing a specific subset of T-cells (MAIT cells) to identify possible roles for the identified metabolites in colonization resistance. Both groups of mice are treated with Cefoperazone antibiotics and are followed longitudinally. The authors build on a previously published manuscript where the microbiome composition of the same mice has been analyzed.
In principle the data that is presented is interesting. However, there data is currently not structured logically are there are some issues with how it is presented. I have the following suggestions to make the data suitable for publication.
My main concerns:
- Very method-heavy abstract and conclusion. There should be a clearer separation between the sections.
- Related to point 1, the results are simply a description of the data that is also plotted. Would a results+discussion format be more appropriate?
- The authors should make it more clear how MAIT cells fit into the general picture of mucosal immunology and why this data is important
- The main question that arises with me is why the KO and WT mice are so different in their response to antibiotics. Possible explanations using the evidence that is generated should be added.
- Flow of presentation of the data is strange. Why start the results with this section? I advise first comparing metabolome profiles of WT and KO at baseline. In the next separate section add the antibiotics and provide rationale for this.
- Boxplots cannot be used to plot this data since they are only suitable for larger group sizes >5. This has to be changed.
- Tables should be moved to supplement and heatmap or similar presentation brought to the main text.
- Authors should provide a link to all the metabolomics data in the main text.
Author Response
- Very method-heavy abstract and conclusion. There should be a clearer separation between the sections.
We removed some study details from the abstract and made some format change in Section “5. Conclusion”. Hope the separation is clear.
- Related to point 1, the results are simply a description of the data that is also plotted. Would a results+discussion format be more appropriate?
The authors appreciate the reviewer’s comment. However, the authors are using the Metabolites format and think the current format might be easier for the readers to understand the work.
- The authors should make it more clear how MAIT cells fit into the general picture of mucosal immunology and why this data is important
Please see Response to the last comment of Review #2.
- The main question that arises with me is why the KO and WT mice are so different in their response to antibiotics. Possible explanations using the evidence that is generated should be added.
Figure 5 and the first paragraph of Discussion can provide you some explanation.
- Flow of presentation of the data is strange. Why start the results with this section? I advise first comparing metabolome profiles of WT and KO at baseline. In the next separate section add the antibiotics and provide rationale for this.
This comment is confusing. Results section starts with “Riboflavin and bacteria-biosynthesized riboflavin metabolites levels” because it is related to the main topic of the study - MR1-/-. Total BA and individual BA are then discussed. Thank the reviewer’s comment regarding the order of baseline and response to the Cef treatments description in the writing. The authors have different opinion since each plot of Figure 1-4 contains both baseline data and data after treatments, which might be easier for the readers to follow if data is described in the order of before and after treatments.
- Boxplots cannot be used to plot this data since they are only suitable for larger group sizes >5. This has to be changed.
The authors agree with the Reviewer’s comment. Box plots have been changed to Bar plots in the manuscript.
- Tables should be moved to supplement and heatmap or similar presentation brought to the main text.
Done!
- Authors should provide a link to all the metabolomics data in the main text.
Other metabolomics findings will be published separately. Please see the Response to the first Comment of Reviewer #2.
Reviewer 2 Report
In this study, the authors profile the composition of bile acids in MR1 knockout mice compared with wild-type mice over a 5-day course of treatment with the antibiotic Cefoperazone. Mouse cecal, intestinal, and fecal samples were analyzed using untargeted mass spectrometry, finding complex shifts over the treatment course that differed substantially in the knockout mice. This study describes an interesting dataset, but the findings are presented confusingly and redundantly, while incompletely reporting some crucial details. Comments: -Why does this study only report findings for riboflavin metabolites, bile acids, and taurine? If untargeted LC-MS was used as described, presumably many other features were also detected. Basic summary statistics, quality checks, and processing for the whole dataset should at least be fully reported, even if additional analyses of other metabolites are planned for another publication. Similarly, the methods section notes that bile acid metabolites were identified based on library standards, but the method used to identify the riboflavin metabolites and taurine is not specified. -HMDB needs to be cited correctly in the methods, see https://hmdb.ca/citing. -Boxplots are not helpful visualizations when there are only 3 data points. The authors should just plot the points themselves. -Log feature intensities are reported in many places, but the log base is not specified. -The tables show redundant information with Figure 3 (although Figure 3 includes only a subset). I recommend removing the tables and simply including the full metabolite dataset as a supplementary dataset. -Adding up the intensities across distinct bile acids to calculate total BA is not statistically meaningful or necessarily reflective of their total concentration, as is implied. The composition of bile acids is changing and some may have relatively higher MS signals than others. Instead, log intensities could be scaled to z-scores and their average enrichment or depletion relative to a baseline sample could be calculated across multiple metabolites. -The hypothesized relationship between bile acids, MAIT cells /knockout status, and antibiotic treatment over time is not clear. A model or schematic figure in the discussion that illustrates the interactions between these variables that are supported by these new findings would be a valuable addition to this study.Author Response
Reviewer #2
In this study, the authors profile the composition of bile acids in MR1 knockout mice compared with wild-type mice over a 5-day course of treatment with the antibiotic Cefoperazone. Mouse cecal, intestinal, and fecal samples were analyzed using untargeted mass spectrometry, finding complex shifts over the treatment course that differed substantially in the knockout mice. This study describes an interesting dataset, but the findings are presented confusingly and redundantly, while incompletely reporting some crucial details. Comments: -Why does this study only report findings for riboflavin metabolites, bile acids, and taurine? If untargeted LC-MS was used as described, presumably many other features were also detected. Basic summary statistics, quality checks, and processing for the whole dataset should at least be fully reported, even if additional analyses of other metabolites are planned for another publication.
Response: We agree with the reviewer’s comments, many metabolic features (especially bacteria-related metabolites) were observed. We are planning to have a separate report for the rest of the data in the future. Line 218-227 in the Discussion “In order to assure the metabolomics data quality, the retention time shift and the mass accuracy were monitored during the run. The retention time shifts of the 40 compounds present in the synthetic... “ was added. In addition, one reference (J Chromatogr B Analyt Technol Biomed Life Sci 2016, 1008:15-25) regarding the synthetic QC sample was added.
Current manuscript reports the findings for bile acids, riboflavin and taurine. Line 21-22 in the Abstract “Since MAIT cell activation depends on riboflavin intermediate, rRL-6-CH2OH (an MR1-restricted riboflavin derivative) was also evaluated.” explains why riboflavin data is reported. Line 194-196 “Changes in the levels of free taurine were linked to the levels of the taurine-conjugated BAs and diminished unconjugated BAs….” explains why taurine is reported. Line 61-72 in the Introduction, Line 119-128 in the Results, and almost the whole Discussion explains why bile acid data is reported. In short summary, 1) bile acids have been reported to be linked to the Clostridioides difficile (CD) germination and growth; 2) MR1 knockout mouse model showed CD infection resistance in our previous study, 3) bile acids are observed dramatically different prior and after antibiotic treatments in the KO mice vs WT. As such, we report the bile acid data to provide new information regarding the MR1 KO mouse model.
Similarly, the methods section notes that bile acid metabolites were identified based on library standards, but the method used to identify the riboflavin metabolites and taurine is not specified. -HMDB needs to be cited correctly in the methods, see https://hmdb.ca/citing.
Citation has been changed as suggested.
Line 278 and 322, “riboflavin and taurine” was added.
Line 324 “The identity of rRL-6-CH2OH was confirmed by the accurate mass and the fragmentation spectrum as published by others [2].” was added.
-Boxplots are not helpful visualizations when there are only 3 data points. The authors should just plot the points themselves.
Box plots have been changed to Bar plots in the manuscript.
-Log feature intensities are reported in many places, but the log base is not specified.
Change log to loge through out of the manuscript.
-The tables show redundant information with Figure 3 (although Figure 3 includes only a subset). I recommend removing the tables and simply including the full metabolite dataset as a supplementary dataset.
The Tables were moved to Supplemental material as suggested.
-Adding up the intensities across distinct bile acids to calculate total BA is not statistically meaningful or necessarily reflective of their total concentration, as is implied. The composition of bile acids is changing and some may have relatively higher MS signals than others. Instead, log intensities could be scaled to z-scores and their average enrichment or depletion relative to a baseline sample could be calculated across multiple metabolites.
The authors agree with the reviewer’s concern that the bile acids have different ionization efficiency. As such, adding BA intensities might not reflect total BA concentrations. We must agree that it is very common to summing the molecule intensities in the analytical chemistry field when internal standards are not available (as exampled Am J Physiol Endocrinol Metab 306: E854–E868, 2014). However, Figure S1 based on log intensity can give such information about enrichments or depletion of individual BA vs baseline. In our work, we are also interested in comparing baseline BA levels in KO vs WT. Therefore, intensity in the bar graph is used for analysis.
-The hypothesized relationship between bile acids, MAIT cells /knockout status, and antibiotic treatment over time is not clear. A model or schematic figure in the discussion that illustrates the interactions between these variables that are supported by these new findings would be a valuable addition to this study.
Figure 5 was added on Pg 6 as suggested.
Reviewer 3 Report
This is a very fine pilot study with sound methods and reasonable conclusions. It builds on prior work by the same group and further elaborates on the bile acid abundance and profile differences that may explain relative resistance to C.difficile colonization.
I wonder if exchanging Tables 1-3 for the heatmaps in Figure S1 would make the manuscript shorter and gentler to a general audience.
For a future study it would be of interest to see correlations between relative abundances of specific OTUs and the BA intensity changes. Also, the use of a well-validated CDI mouse model such as (doi:10.1053/j.gastro.2008.09.002) would increase the interest.
Author Response
Reviewer#3
This is a very fine pilot study with sound methods and reasonable conclusions. It builds on prior work by the same group and further elaborates on the bile acid abundance and profile differences that may explain relative resistance to C.difficile colonization.
I wonder if exchanging Tables 1-3 for the heatmaps in Figure S1 would make the manuscript shorter and gentler to a general audience.
Thank you for the comment. The Tables were moved to the Supplemental Material.
For a future study it would be of interest to see correlations between relative abundances of specific OTUs and the BA intensity changes. Also, the use of a well-validated CDI mouse model such as (doi:10.1053/j.gastro.2008.09.002) would increase the interest.
It is our future plan to integrate metagenomics data with metabolomics data, although this manuscript reported some microbiome data (for instance changes in Clostridiales, responsible for the dehydroxylation reactions in the large intestine, was consistent with DCA changes) were consistent with some bile acid data.
Round 2
Reviewer 1 Report
Comments on revised version.
Barplots are an improvement but why not plot the actual datapoints on them as is common these days?
Why not bring the supplementary heatmap into the main text and do bi-directional hierarchical clustering on it?
I do not understand the comment made on the order of presenting the data but if the authors want to keep it like this that is OK with me.
Author Response
Reviewer#1:
Comments on revised version.
Barplots are an improvement but why not plot the actual datapoints on them as is common these days?
Figures have been changed as suggested.
Why not bring the supplementary heatmap into the main text and do bi-directional hierarchical clustering on it?
The authors appreciate the reviewer’s suggestion. The supplemental heatmap can only give the relative comparison between the treated groups vs D0; however, it is unable to show the relative semi-quantitative difference between the KO vs WT. The barplots with individual animal data (Figure 4) can show more information including the changes between the treated vs D0, difference between KO vs WT, as well as individual animal data distributions. As such, Figure 4 is chosen in the main text.
I do not understand the comment made on the order of presenting the data but if the authors want to keep it like this that is OK with me.
Thank you.
Reviewer 2 Report
I thank the authors for their responses and revisions. I have two minor comments:
-If the analysis of total bile acid intensities is retained, it is important to always refer to this quantity as "total bile acid signal" or "total bile acid intensities" rather than "total bile acids", which implies concentration. "Total biles acids" or "total BAs" is used in the abstract and a few other places.
-The bar plots would be improved if the individual raw data points were plotted on top of the bars to illustrate the number and variability of measurements in each group.
Author Response
Reviewer#2:
I thank the authors for their responses and revisions. I have two minor comments:
-If the analysis of total bile acid intensities is retained, it is important to always refer to this quantity as "total bile acid signal" or "total bile acid intensities" rather than "total bile acids", which implies concentration. "Total biles acids" or "total BAs" is used in the abstract and a few other places.
“Total bile acids” and “total BAs” have been changed to “total bile acid intensities” throughout the manuscript.
-The bar plots would be improved if the individual raw data points were plotted on top of the bars to illustrate the number and variability of measurements in each group.
Figures have been changed as suggested.